# RIESZNET AND FORESTRIESZ:
# AUTOMATIC DEBIASED MACHINE LEARNING WITH NEURAL NETS AND RANDOM FORESTS

## ABSTRACT

Many causal and policy effects of interest are defined by linear functionals of high-dimensional or non-parametric regression functions. $\sqrt{n}$-consistent and asymptotically normal estimation of the object of interest requires debiasing to reduce the effects of regularization and/or model selection on the object of interest. Debiasing is typically achieved by adding a correction term to the plug-in estimator of the functional, that is derived based on a functional-specific theoretical derivation of what is known as the influence function and which leads to properties such as double robustness and Neyman orthogonality. We instead implement an automatic debiasing procedure based on automatically learning the Riesz representation of the linear functional using Neural Nets and Random Forests. Our method solely requires value query oracle access to the linear functional. We propose a multi-tasking Neural Net debiasing method with stochastic gradient descent minimization of a combined Riesz represented and regression loss, while sharing representation layers for the two functions. We also propose a Random Forest method which learns a locally linear representation of the Riesz function. Even though our methodology applies to arbitrary functionals, we experimentally find that it beats state of the art performance of the prior neural net based estimator of Shi et al. (2019) for the case of the average treatment effect functional. We also evaluate our method on the more challenging problem of estimating average marginal effects with continuous treatments, using semi-synthetic data of gasoline price changes on gasoline demand.

## 1 INTRODUCTION

A large number of problems in causal inference, off-policy evaluation and optimization and interpretable machine learning can be viewed as estimating the average value of a moment function that depends on an unknown regression function:

$$\theta_0 = \mathbb{E}[m(W; g_0)], \quad \text{where } W := (Y, Z) \text{ and } g_0(Z) := \mathbb{E}[Y \mid Z].$$

In most cases, $Y$ will be the outcome of interest, and inputs $Z = (T, X)$ will include a binary or continuous treatment $T$ and covariates $X$. Prototypical examples include the estimation of average treatment effects, average policy effects, average derivatives and incremental policy effects.

**Example 1** (Average treatment effect). *Here $Z = (T, X)$ where $T$ is a binary treatment indicator, and $X$ are covariates. The object of interest and associated moment function are:*

$$\theta_0 = \mathbb{E}[g_0(1, X) - g_0(0, X)], \quad m(W; g) = g(1, X) - g(0, X).$$

*If potential outcomes are conditionally independent of treatment $T$ given covariates $X$, then this object is the average treatment effect (Rosenbaum & Rubin, 1983).*

**Example 2** (Average policy effect). *In the context of offline policy evaluation and optimization, our goal is to optimize over a space of assignment policies $\pi : X \to \{0, 1\}$, when having access to observational data collected by some unknown treatment policy. The policy value can also be formulated as the average of a linear moment:*

$$\theta_0 = \mathbb{E}[\pi(X)(g_0(1, X) - g_0(0, X)) + g_0(0, X)], \quad m(W; g) = \pi(X)(g(1, X) - g(0, X)) + g(0, X).$$

*A long-line of prior work has considered doubly-robust approaches to optimizing over a space of candidate policies from observational data (see e.g. Dudík et al., 2011; Athey & Wager, 2021).*

**Example 3** (Average marginal effect / derivative). *Here $Z = (T, X)$, where $T$ is a continuously distributed policy variable of interest, $X$ are covariates, and:*

$$\theta_0 = \mathbb{E}\left[\frac{\partial g_0(T, X)}{\partial t}\right], \quad m(W; g) = \frac{\partial g(T, X)}{\partial t}.$$

*This object is essentially the average slope in the partial dependence plot frequently used in work on interpretable machine learning (see e.g. Zhao & Hastie, 2021; Friedman, 2001; Molnar, 2020).*

**Example 4** (Incremental policy effects). *Here $Z = (T, X)$, where $T$ is a continuously distributed policy variable of interest, $X$ are covariates, and $\pi : X \to [-1, 1]$ is an incremental policy of infinitesimally increasing or descreasing the treatment from its baseline value (see e.g. Athey & Wager, 2021). The incremental value of such an infinitesimal policy change takes the form:*

$$\theta_0 = \mathbb{E}\left[\pi(X)\frac{\partial g_0(T, X)}{\partial t}\right], \quad m(W; g) = \pi(X)\frac{\partial g(T, X)}{\partial t}.$$

*Such incremental policy effects can also be useful within the context of policy gradient algorithms in deep reinforcement learning, so as to take gradient steps towards a better policy, and debiasing techniques have already been used in that context (Grathwohl et al., 2017).*

Even though the non-parametric regression function is typically estimable only at slower than parametric rates, one can often achieve parametric rates for the average moment function. However, this is typically not achieved by simply pluging a non-parametric regression estimate into the moment formula and averaging, but requires debiasing approaches to reduce the effects of regularization when learning the non-parametric regression.

Typical debiasing techniques are tailored to the moment of interest. In this work we present automatic debiasing techniques that use the representation power of neural nets and random forests and which only require oracle access to the moment of interest. Our resulting average moment estimators are typically consistent at parametric $\sqrt{n}$ rates and are asymptotically normal, allowing for the construction of confidence intervals with approximately nominal coverage. The latter is essential in social science applications and can also prove useful in policy learning applications, so as to quantify the uncertainty of different policies and implement automated policy optimization algorithms which require uncertainty bounds (e.g. algorithms that use optimism in the face of uncertainty).

Relative to previous works in the automatically debiased ML (Auto-DML) literature, the contribution of this paper is twofold. On the one hand, we provide the first practical implementation of Auto-DML using neural networks (RieszNet) and random forests (ForestRiesz). As such, we complement the theoretical guarantees of Chernozhukov et al. (2021) for generic machine learners. On the other hand, we show that our methods perform better than existing benchmarks and that inference based on asymptotic confidence intervals obtains coverage close to nominal in two settings of great relevance in applied research: the average treatment effect of a binary treatment and the average marginal effect (derivative) of a continuous treatment.

The rest of the paper is structured as follows. Section 2 provides some background on estimation of average moments of regression functions. In Section 2.1 we describe the form of the debiasing term, and in Section 2.2 we explain how it can be automatically estimated. Sections 3 and 4 introduce our proposed estimators: RieszNet and ForestRiesz, respectively. Finally, in Section 5 we present our experimental results.

## 2 ESTIMATION OF AVERAGE MOMENTS OF REGRESSION FUNCTIONS

### 2.1 DEBIASING THE AVERAGE MOMENT

We focus on problems where there exists a function $\alpha_0(Z)$ with finite second moment such that:

$$\mathbb{E}[m(W; g)] = \mathbb{E}[\alpha_0(Z)g(Z)], \quad \text{for all } g \text{ with } \mathbb{E}[g(Z)^2] < \infty.$$

By the Riesz representation theorem, existence of such an $\alpha_0(X)$ is equivalent to $\mathbb{E}[m(W; g)]$ being a mean-square continuous linear functional of $g$. We will refer to this $\alpha_0(X)$ as the Riesz representer (RR). Existence of the RR is equivalent to the semiparametric variance bound for $\theta_0$ being finite, as in Newey (1994) and Hirshberg and Wager (2018).

This RR exists in each of Examples 1 to 4. For instance, in Example 1 the RR is $\alpha_0(Z) = T/p_0(X) - (1-T)/(1-p_0(X))$ where $p_0(X) = \Pr(T = 1 \mid X)$ is the propensity score and in Example 3, integration by parts gives $\alpha_0(Z) = -\left(\partial f_0(T, X)/\partial t\right)/f_0(Z)$ where $f_0(Z)$ is the joint probability density function (pdf) of $T$ and $Z$. In general, the RR involves (unknown) nonparametric functions of the data, like the propensity score or the density $f_0(Z)$ and its derivative.

The RR is a crucial object in the debiased ML literature, since it allows us to construct a debiasing term for the moment function $m(W; g)$ (see Chernozhukov et al., 2018a, for details). The debiasing term in this case takes the form $\alpha(Z)(Y - g(Z))$. To see that, consider the score $m(W; g) + \alpha(Z)(Y - g(Z)) - \theta$. It satisfies the following property:

$$\mathbb{E}[m(W; g) + \alpha(Z)(Y - g(Z)) - \theta_0] = -\mathbb{E}[(\alpha(Z) - \alpha_0(Z))(g(Z) - g_0(Z))].$$

This is sometimes known as *double robustness*, since the score will be zero in expectation when either $\alpha(Z) = \alpha_0(Z)$ or $g(Z) = g_0(Z)$ are correctly specified. An estimator of $\theta_0$ can be constructed from this score and first-stage estimators $\widehat{g}$ and $\widehat{\alpha}$. Let $\mathbb{E}_n[\cdot]$ denote the empirical expectation over a sample of size $n$, i.e. $\mathbb{E}_n[Z] = \frac{1}{n}\sum_{i=1}^{n} Z_i$. We consider:

$$\widehat{\theta} = \mathbb{E}_n\left[m(W; \widehat{g}) + \widehat{\alpha}(Z)(Y - \widehat{g}(Z))\right]. \tag{1}$$

What double robustness means in practice is that the bias of this estimator will vanish at a rate equal to the product of the mean-square convergence rates of $\widehat{\alpha}$ and $\widehat{g}$. Therefore, in cases where the regression function $g_0$ can be estimated very well, the rate requirements on $\widehat{\alpha}$ will be less strict, and vice versa. More notably, whenever the product of the mean-square convergence rates of $\widehat{\alpha}$ and $\widehat{g}$ is larger than $\sqrt{n}$, the estimator $\widehat{\theta}$ will be asymptotically normal at the parametric rate $\sqrt{n}$, as proven formally in Theorem 4 of Chernozhukov et al. (2021).

The regression estimator $\widehat{g}$ and the RR estimator $\widehat{\alpha}$ may use samples different than the $i$-th, which constitutes cross-fitting. Cross-fitting reduces bias from using the $i$-th sample in estimating $\alpha$ and $g$. Also $\widehat{g}$ and $\widehat{\alpha}$ may use different samples, which constitutes double cross-fitting (see e.g. Newey & Robins, 2018, for the benefits of double cross-fitting in reducing the requirements on the quality of the regression and RR estimates).

## 2.2 RIESZ REPRESENTER AS MINIMIZER OF STOCHASTIC LOSS

The theoretical foundation for this paper is the recent work of Chernozhukov et al. (2021), who show that one can view the Riesz representer as the minimizer of the loss function:

$$\alpha_0 = \underset{\alpha \in \mathcal{A}_n}{\arg\min}\, \mathbb{E}[\alpha(Z)^2 - 2m(W; \alpha)]$$

over some sufficiently flexible hypothesis space $\mathcal{A}$ and hence consider an empirical estimate of the Riesz representer by minimizing the corresponding empirical loss within some hypothesis space $\mathcal{A}$:

$$\widehat{\alpha} = \underset{\alpha \in \mathcal{A}_n}{\arg\min}\, \mathbb{E}_n[\alpha(Z)^2 - 2m(W; \alpha)] \tag{2}$$

The benefits of estimating the RR using this loss are twofold: (i) we do not need to derive an analytic form of the RR of the object of interest, (ii) we are trading-off bias and variance for the actual RR, since the loss is asymptotically equivalent to the square loss $\mathbb{E}[(\alpha_0(Z) - \alpha(Z))^2]$, as opposed to plug-in Riesz estimators that first solve some classification, regression or density estimation problem and then plug the resulting estimate into the analytic RR formula. This approach can lead to finite sample instabilities, for instance, in the case of binary treatment effects, when the propensity scores are close to 0 or 1 and they appear in the denominator of the RR. Prior work by Chernozhukov et al. (2018b) optimized the loss function in equation 2 over linear Riesz functions with a growing feature map, while Chernozhukov et al. (2020) allowed for the estimation of the RR in arbitrary function spaces, but proposed a computationally harder minimax loss formulation.

From a theoretical standpoint, Chernozhukov et al. (2021) also provide fast statistical estimation rates. Let $\| \cdot \|_2$ denote the $\ell_2$ norm of a function of a random input, i.e. $\|\alpha\|_2 = \sqrt{\mathbb{E}[\alpha(Z)^2]}$. We also let $\| \cdot \|_\infty$ denote the $\ell_\infty$ norm, i.e. $\|a\|_\infty = \max_{z \in \mathcal{Z}} |a(z)|$.

**Theorem 1** (Chernozhukov et al. (2021)). *Let $\delta_n$ be an upper bound on the critical radius (Wainwright, 2019) of the function spaces:*

$$\{z \mapsto \gamma\,(\alpha(z) - \alpha_0(z)) : \alpha \in \mathcal{A}_n, \gamma \in [0,1]\}$$
$$\textit{and } \{w \mapsto \gamma\,(m(w; \alpha) - m(w; \alpha_0)) : \alpha \in \mathcal{A}_n, \gamma \in [0,1]\}, \tag{3}$$

*and suppose that for all $f$ in the spaces of Equation* (3): $\|f\|_\infty \le 1$. *Suppose, furthermore, that $m$ satisfies the mean-squared continuity property:*

$$\forall \alpha, \alpha' \in \mathcal{A} : \mathbb{E}[(m(W; \alpha) - m(W; \alpha'))^2] \le M \|\alpha - \alpha'\|_2^2$$

*for some $M \ge 1$. Then for some universal constant $C$, we have that w.p. $1 - \zeta$:*

$$\|\widehat{\alpha} - \alpha_0\|_2^2 \le C\left(\delta_n^2\,M + \frac{M \log(1/\zeta)}{n} + \inf_{\alpha_* \in \mathcal{A}} \|\alpha_* - \alpha_0\|_2^2\right) \tag{4}$$

The critical radius is a quantity that has been analyzed for several function spaces of interest, such as high-dimensional linear functions with bounded norms, neural networks and shallow regression trees, many times showing that $\delta_n = O(d_n\,n^{-1/2})$, where $d_n$ are various notions of dimension of the hypothesis space (see e.g. Chernozhukov et al., 2021, for concrete rates). Theorem 1 can be applied to provide fast statistical estimation guarantees for the corresponding function spaces, albeit this result does not give guidelines on how to optimize over such function spaces in practice. In our work, we provide heuristics for taking this theorem to practice for the case of neural networks and random forests and propose several practical enhancements.

## 3 RieszNet: Targeted Regularization and Multi-tasking

Our design of the RieszNet architecture starts by showing the following lemma:

**Lemma 1.** *In order to estimate the average moment of the regression function $g_0(Z) = \mathbb{E}[Y \mid Z]$ it suffices to estimate regression functions of the form $g_0(Z) = h_0(\alpha_0(Z))$, where $h_0(A) = \mathbb{E}[Y \mid A]$ and $A = \alpha_0(Z)$ is the evaluation of the Riesz representer at a sample. In other words, it suffices to estimate a regression function that solely conditions on the value of the Riesz representer.*

*Proof.* It is easy to verify that:

$$\theta_0 = \mathbb{E}[m(W; g_0)] = \mathbb{E}[g_0(Z)\alpha_0(Z)] = \mathbb{E}[Y\alpha_0(Z)]$$
$$= \mathbb{E}[\mathbb{E}[Y \mid A = \alpha_0(Z)]\alpha_0(Z)] = \mathbb{E}[h_0(\alpha_0(Z))\alpha_0(Z)] = \mathbb{E}[m(W; h_0 \circ \alpha_0)]. \qquad \square$$

This property is a generalization of the observation that, in the case of average treatment effect estimation, it suffices to condition on the propensity value of each sample and the treatment value (Rosenbaum & Rubin, 1983). In the case of the average treatment effect moment, these two quantities suffice to reproduce the Riesz representer. The aforementioned observation generalizes this well-known fact in causal estimation, which was also invoked in the prior work of Shi et al. (2019).

Lemma 1 allows us to argue that, when estimating the regression function, it suffices to use features that are predictive of the Riesz representer. This leads to a multi-tasking neural network architecture, which is a generalization of that of Shi et al. (2019) to arbitrary linear functionals and moment functions.

We consider a deep neural representation of the RR of the form: $\alpha_0(Z; \beta, w_{1:k}) = \langle \beta, f_1(Z; w_{1:k})\rangle$, where $f_1(X; w_{1:k})$ is the final feature representation layer of an arbitrary deep neural architecture with $k$ hidden layers and weights $w_{1:k}$. The goal of the Riesz estimate is to minimize the Riesz loss:

$$\text{RRloss}(\beta, w_{1:k}) := \mathbb{E}_n\left[\alpha(Z; \beta, w_{1:k})^2 - 2\,m(W; \alpha(\cdot; \beta, w_{1:k}))\right] \tag{5}$$

In the limit, the representation layer $f_1(Z; w_{1:k})$ will contain sufficient information to represent the true RR. Thus, conditioning on this layer to construct the regression function, suffices to get a consistent estimate. Hence, even if these features are completely driven by predicting the RR, they will be a super-set that is required by Lemma 1.

Based on this observation, we will represent the regression function with a deep neural network, starting from the final layer of the Riesz representer, i.e. $g(Z; w_{1:d}) = f_2(f_1(Z; w_{1:k}); w_{(k+1):d})$,

with $d - k$ additional hidden layers and weights $w_{(k+1):d}$. The regression is simply trying to minimize the square loss:

$$\text{REGloss}(w_{1:d}) := \mathbb{E}_n \left[ (Y - g(Z; w_{1:d}))^2 \right] \tag{6}$$

Note that the parameters of the common layers also enter the regression loss, and hence even if the RR function is a constant, their gradient information will primarily be driven by the regression loss and will reduce variance by explaining more of the output $Y$.

Finally, we will add a regularization term that is the analogue of the targeted regularization introduced by Shi et al. (2019). In fact, the intuition behind the following regularization term dates back to the early work of Bang & Robins (2005). Bang & Robins (2005) observed that one can show double robustness of a plug-in estimator in the case of estimation of average effects, if one simply adds the inverse propensity as a regression variable, in a linear manner, and does not penalize its coefficient. This idea generalizes to Riesz functions and general moments. In particular, if we add the RR as an extra input to our regression problem in a linear manner, i.e. learn a regression function of the form: $\tilde{g}(Z) = g(Z; w_{1:d}) + \epsilon \cdot \alpha_0(Z)$, where $\epsilon$ is an un-penalized parameter. Then note that if we minimize the square loss with respect to $w_{1:d}$ and $\epsilon$, then the resulting estimate, will satisfy the property (due to the first order condition with respect to $\epsilon$), that:

$$\mathbb{E}_n \left[ (Y - g(Z; w_{1:d}) - \epsilon \cdot \alpha_0(Z)) \cdot \alpha_0(Z) \right] = 0$$

Then note that the debiasing correction in the doubly-robust moment formulation is identically equal to zero when we use the regression function $\tilde{g}$, since: $\mathbb{E}_n \left[ (Y - \tilde{g}(Z)) \cdot \alpha_0(Z) \right] = 0$. Thus the plug-in estimate of the average moment is equivalent to the doubly-robust estimate, when one uses the regression model $\tilde{g}$, since:

$$\widehat{\theta} = \mathbb{E}_n[m(Z; \tilde{g})] = \mathbb{E}_n[m(Z; \tilde{g})] + \mathbb{E}_n \left[ (Y - \tilde{g}(Z)) \cdot \alpha_0(Z) \right]$$

A similar intuition underlines the TMLE framework. However, in that framework, the parameter $\epsilon$ is not simultaneously optimized together with the regression parameters $w$, but rather in a post-processing step: first an arbitrary regression model $g$ is fitted (via any regression approach) and subsequently the preliminary $g$ is corrected by solving a linear regression problem between the residuals $Y - g(Z)$ and the Riesz representer $\alpha(Z)$, to estimate a coefficient $\epsilon$, i.e. minimizing the square loss:

$$\mathbb{E}_n \left[ (Y - g(Z) - \epsilon \cdot \alpha(Z))^2 \right].$$

Then the corrected regression model $g(Z) + \epsilon \cdot \alpha(Z)$ is used in a plug-in manner to estimate the average moment. For an overview of these variants of doubly-robust estimators see Tran et al. (2019). In that respect, our Riesz estimation approach can be viewed as automating the process of identifying the least favorable parametric sub-model required by the TMLE framework and which is typically done on a case-by-case basis and based on analytical derivations of the efficient influence function and contributes to the recent line of work on such automated TMLE (Carone et al., 2019).

In this work, similar to Shi et al. (2019) we take an intermediate avenue, where the correction regression loss from the TMLE post-processing step is added as a regularization term, leading to the overall loss that is optimized by our multi-tasking deep architecture:

$$\min_{\beta, w_{1:d}, \epsilon} \text{RRloss}(\beta, w_{1:k}) + \lambda_1 \text{REGloss}(w_{1:d})$$
$$+ \lambda_2 \mathbb{E}_n \left[ (Y - g(Z; w_{1:d}) + \epsilon \cdot \alpha(Z; \beta, w_{1:k}))^2 \right] + R(\beta, w_{1:d}) \tag{7}$$

where $R$ is any regularization penalty on the parameters of the neural network, which crucially does not take $\epsilon$ as input. Minimizing the neural network parameters of the loss defined in Equation (7) using stochastic first order methods constitutes our RieszNet estimation method for the average moment of a regression function. Note that in the extreme case when $\lambda_1 = 0$, then the second loss is equivalent to the one-step approach of Bang & Robins (2005), while as $\lambda_2$ goes to zero the parameters $w_{1:d}$ are primarily optimized based on the square loss, and hence the $\epsilon$ is estimated given a fixed regression function $g$, thereby mimicking the two-step approach of the TMLE framework.

## 4 FORESTRIESZ: LOCALLY LINEAR RIESZ ESTIMATION

One approach to constructing a tree that approximates the solution to the Riesz loss minimization problem is to simply use the Riesz loss as a criterion function when finding an optimal split among

all variables $Z$. However, we note that this approach introduces a large discontinuity in the treatment variable $T$, which is part of $Z$. Such discontinuous in $T$ function spaces will typically not satisfy the mean-squared continuity property. Furthermore, since the moment function typically evaluates the function input at multiple treatment points, the critical radius of the resulting function space $m \circ \alpha$ runs the risk of being extremely large and hence the estimation error not converging to zero. Moreover, unlike the case of a regression forest, it is not clear what the "local node" solution will be if we are allowed to split on the treatment variable, since the local minimization problem can be ill-posed.

As a concrete case, consider the example of an average treatment effect of a binary treatment. One could potentially minimize the Riesz loss by constructing child nodes that contain no samples from one of the two treatments. In that case the local node solution to the Riesz loss minimization problem is not well-defined.

For this reason, we consider an alternative formulation, where the tree is only allowed to split on variables other than the treatment, i.e. the variables $X$. Then we consider a representation that is locally linear with respect to some pre-defined feature map $\phi(T, X) \in \mathbb{R}^d$ (e.g. a polynomial series). Then the Riesz function is represented in the form: $\alpha(Z) = \beta(X) \cdot \phi(T, X)$, where $\beta(X)$ is a non-parametric (potentially discontinuous) function estimated based on the tree splits and $\phi(T, X)$ is a smooth feature map. In that case, by the linearity of the moment, the Riesz loss takes the form:

$$\min_\beta \mathbb{E}_n[\beta(X)\phi(Z)\phi(Z)'\beta(X) - 2\,\beta(X)'m(W; \phi)] \tag{8}$$

where we use the short-hand notation $m(W; \phi) = (m(W; \phi_1), \ldots, m(W; \phi_d))$. Since $\beta(\cdot)$ is allowed to be fully non-parametric, we can equivalently formulate the above minimization problem as satisfying the local first-order conditions conditional on each target $x$, i.e.:

$$\beta(x) \text{ solves}: \quad \mathbb{E}[\phi(Z)\phi(Z)'\beta(x) - m(W; \phi) \mid X = x] = 0 \tag{9}$$

This problem falls in the class of problems defined via solutions to moment equations. Hence, we can apply the recent framework of Generalized Random Forests of Athey et al. (2019) to solve this local moment problem via random forests.

That is exactly the approach we take in this work. We note that we depart from the exact algorithm presented in Athey et al. (2019) in that we slightly modify the criterion function to not solely maximize the heterogeneity of the resulting local estimates from a split (as in Athey et al., 2019), but rather to exactly minimize the Riesz loss criterion. The two criteria are slightly different. In particular, when we consider the splitting of a root node into two child nodes, then Athey et al. (2019) chooses a split that maximizes $N_1\beta_1(X)^2 + N_2\beta_2(X)^2$. Our criterion penalizes splits where the local jacobian matrix:

$$J(\text{child}) := \frac{1}{|\text{child}|} \sum_{i \in \text{child}} \phi(Z_i)\phi(Z_i)'$$

is not ill-posed (where $|\text{child}|$ denotes the number of samples in a child node). In particular, note that the local solution at every leaf is of the form:

$$\beta(\text{child}) = J(\text{child})^{-1}M(\text{child}) \qquad M(\text{child}) := \frac{1}{|\text{child}|} \sum_{i \in \text{child}} m(W_i; \phi) \tag{10}$$

and the average Riesz loss after a split is proportional to:

$$-\sum_{\text{child} \in \{1,2\}} |\text{child}|\,\beta(\text{child})'J(\text{child})\beta(\text{child}).$$

Hence, minimizing the Riesz loss is equivalent to maximizing the negative of the above quantity. Note that the heterogeneity criterion of Athey et al. (2019) would simply maximize $\sum_{\text{child} \in \{1,2\}} |\text{child}|\,\beta(\text{child})'\beta(\text{child})$, ignoring the ill-posedness of the local Jacobian matrix. However, we note that the consistency results of Athey et al. (2019) do not depend on the exact criterion that is used and solely depend on the splits being sufficiently random and balanced. Hence, they easily extend to the criterion that we use here.

Finally, we note that our forest approach is also amenable to multi-tasking, since we can add to the moment equations the extra set of moment equations that correspond to the regression problem i.e. simply $\mathbb{E}[Y - g(x) \mid X = x] = 0$ and invoking a Generalized Random Forest for the super-set of these equations and the Riesz loss moment equations. This leads to a multi-tasking forest approach that learns a single forest to represent both the regression function and the Riesz function, to be used for subsequent debiasing of the average moment.

## 5 Experimental Results

In this section, we evaluate the performance of RieszNet and ForestRiesz in two settings that are central in causal and policy estimation: the Average Treatment Effect (ATE) of a binary treatment (Example 1) and the Average Derivative of a continuous treatment (Example 3). Throughout this section, we use RieszNet and ForestRiesz to learn the regression function $g_0$ and RR $\alpha_0$, and compare the following three methods: (i) direct, (ii) Inverse Propensity Score weighting (IPS) and (iii) doubly-robust (DR):

$$\widehat{\theta}_{\text{direct}} = \mathbb{E}_n[m(W; \widehat{g})], \qquad \widehat{\theta}_{\text{IPW}} = \mathbb{E}_n[\widehat{\alpha}(Z)Y],$$
$$\widehat{\theta}_{\text{DR}} = \mathbb{E}_n\left[m(W; \widehat{g}) + \widehat{\alpha}(Z)(Y - \widehat{g}(Z))\right].$$

The first method simply plugs in the regression estimate $\widehat{g}$ into the moment of interest and averages. The second method uses the fact that, by the Riesz representation theorem and the tower property of conditional expectations, $\theta_0 = \mathbb{E}[m(W; g_0)] = \mathbb{E}[\alpha_0(Z)g_0(Z)] = \mathbb{E}[\alpha_0(Z)Y]$. The third, our preferred method, combines both approaches as a debiasing device, as explained in Section 2.

### 5.1 Average Treatment Effect in the IHDP Dataset

Following Shi et al. (2019), we evaluate the performance of our estimators for the Average Treatment Effect (ATE) of a binary treatment on 1000 semi-synthetic datasets based on the Infant Health and Development Program (IHDP). IHDP was a randomized control trial aimed at studying the effect of home visits and attendance at specialized clinics on future developmental and health outcomes for low birth weight, premature infants (Gross, 1993). We use the `NPCI` package in `R` to generate the semi-synthetic datasets under setting "A" (Dorie, 2016). Each dataset consists of 747 observations of an outcome $Y$, a binary treatment $T$ and 25 continuous and binary confounders $X$.

Table 1 presents the mean absolute error (MAE) over the 1000 semi-synthetic datasets. Our preferred estimator, which uses the doubly-robust (DR) moment function to estimate the ATE, achieves a MAE of 0.110 (std. err. 0.003) and 0.126 (std. err. 0.004) when using RieszNet and ForestRiesz, respectively.[1] A natural benchmark against which to compare our Auto-DML methods are plug-in estimators. These use the known form of the Riesz representer for the case of the ATE and an estimate of the propensity score $p_0(X) := \Pr(T = 1 \mid X)$ to construct the Riesz representer as:

$$\widehat{\alpha}(T, X) = \frac{T}{\widehat{p}(X)} - \frac{1 - T}{1 - \widehat{p}(X)}.$$

The state-of-the-art neural-network-based plug-in estimator is the Dragonnet of Shi et al. (2019), which gives a MAE of 0.14 over our 1000 instances of the data. A plug-in estimator where both the regression function and the propensity score are estimated by random forests yields a MAE of 0.389. Hence, automatic debiasing seems a promising alternative to current methods even for causal parameters like the ATE, for which the form of the Riesz representer is well-known.

Table 1: RieszNet and ForestRiesz: Mean Absolute Error (MAE) and its standard error over 1000 semi-synthetic datasets based on the IHDP experiment.

|  (a) RieszNet | MAE ± std. err. |
| --- | --- |
| **DR** | **0.110 ± 0.003** |
| Direct | 0.123 ± 0.004 |
| IPS | 0.122 ± 0.037 |
| **Benchmark:** Dragonnet (Shi et al., 2019) | 0.146 ± 0.010 |

| (b) ForestRiesz | MAE ± std. err. |
| --- | --- |
| **DR** | **0.126 ± 0.004** |
| Direct | 0.197 ± 0.007 |
| IPS | 0.669 ± 0.004 |
| **Benchmark:** RF Plug-in | 0.389 ± 0.024 |

To assess the coverage of our asymptotic confidence intervals in the same setting, we perform another experiment in which this time we also redraw the treatment, according to the propensity score setting "True" in the `NPCI` package. Outcomes are still generated under setting "A."

---

[1]See Appendix A.1 for the architecture and tuning details we used for RieszNet in all experiments.

The results in Figure 1, based on 100 instances of the dataset, show that the performance of RieszNet and ForestRiesz is excellent in terms of coverage when using the doubly-robust (DR) moment. Confidence intervals cover the true parameter 93% and 96% of the time (for a nominal 95%), respectively. The DR moment also has the lowest RMSE. On the other hand, the direct method (which does not use the debiasing term) seems to have lower bias for the RieszNet estimator, although in both cases its coverage is very poor. This is because the standard errors without the debiasing term greatly underestimate the true variance of the estimator.

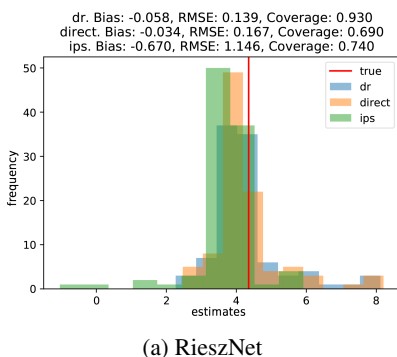

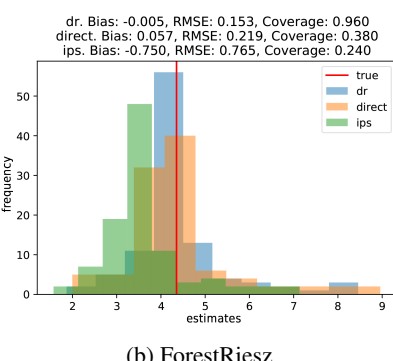

(a) RieszNet

(b) ForestRiesz

Figure 1: RieszNet and ForestRiesz: Bias, RMSE, coverage and distribution of estimates over 100 semi-synthetic datasets based on the IHDP experiment, where we redraw $T$.

## 5.2 BHP GASOLINE DEMAND DATA

To evaluate the performance of our estimators for average marginal effects of a continuous treatment, we conduct a semi-synthetic experiment based on gasoline demand data from Blundell et al. (2017) [BHP]. The dataset is constructed from the 2001 National Household Travel Survey, and contains 3,640 observations at the household level. The outcome of interest $Y$ is (log) gasoline consumption. We want to estimate the effects of changing (log) price $T$, adjusting for differences in confounders $X$, including (log) household income, (log) number of drivers, (log) household respondent age, and a battery of geographic controls.

We generate our semi-synthetic data as follows. First, we estimate $\mu(X) := \mathbb{E}[T \mid X]$ and $\sigma^2(X) :=$ $\mathrm{Var}(T \mid X)$ by a Random Forest of $T$ and $(T - \widehat{\mu}(X))^2$ on $X$, respectively. We then draw 3,640 observations of $T \sim \mathcal{N}(\widehat{\mu}(X), \widehat{\sigma}^2(X))$, and generate $Y = f(T, X) + \epsilon$, for six different choices of $f(\cdot)$. The error term $\epsilon$ is drawn from a $\mathcal{N}(0, \sigma^2)$, with $\sigma^2$ chosen to guarantee that the simulated regression $R^2$ matches the one in the true data.

The exact form of $f$ in each design is detailed in Appendix A.2. In the "simple $f$" designs we have a constant, homogeneous marginal effect of $-0.6$ (within the range of estimates in Blundell et al., 2012, using the real survey data). In the "complex $f$" designs, we have a regression function that is cubic in $T$, and where there are heterogeneous marginal effects by income (built to average approximately $-0.6$). In both cases, we evaluate the performance of the estimators without confounders $X$, and with confounders entering the regression function linearly and non-linearly.

Table 2 presents the results for the most challenging design: a complex regression function with linear and non-linear confounders (see Tables A1 and A2 in the Appendix for full set of results in all designs). ForestRiesz with the doubly-robust moment combined with the TMLE adjustment (in which we use a corrected regression $\widetilde{g}(Z) = \widehat{g}(Z) + \epsilon \cdot \widehat{\alpha}(Z)$, where $\epsilon$ is the OLS coefficient of $Y - \widehat{g}(Z)$ on $\widehat{\alpha}(Z)$) and cross-fitting seems to have the best performance in cases with many linear and non-linear confounders. Both simple cross-fitting with multitasking and double cross-fitting yield coverage close to or above the nominal confidence level (95%), with biases of around one order of magnitude lower than the true effect. As for the ATE, the direct method has low bias but the standard errors underestimate the true variance of the estimator, and so coverage based on asymptotic confidence intervals is poor.[2]

---

[2]As can be seen in the Appendix, ForestRiesz seems to have larger bias and low coverage when there are no confounders compared to RieszNet (both using the IPS or the DR moments).

We can consider a plug-in estimator as a benchmark. Using the knowledge that $T$ is normally distributed conditional on covariates $X$, the plug-in Riesz representer can be constructed using Stein's identity (Lehmann & Casella, 2006), as:

$$\widehat{\alpha}(T, X) = \frac{T - \widehat{\mu}(X)}{\widehat{\sigma}^2(X)},$$

where $\widehat{\mu}(X)$ and $\widehat{\sigma}^2$ are random forest estimates of the conditional mean and variance of $T$, respectively. The results for the plug-in estimator are on Table A3. Surprisingly, we find that our method, which is fully generic and non-parametric, slightly outperforms the plug-in that uses knowledge of the Gaussian conditional distribution.

Table 2: RieszNet and ForestRiesz: bias, RMSE and coverage over 1000 semi-synthetic datasets based on the BHP gasoline price data (10 different random seeds). The DGP is based on a complex regression function with linear and non-linear confounders.

(a) RieszNet

|        | Bias  | RMSE  | Cov.  |
|--------|-------|-------|-------|
| DR     | 0.062 | 0.504 | 0.877 |
| Direct | 0.053 | 0.562 | 0.056 |
| IPS    | 0.061 | 0.496 | 0.916 |

(b) ForestRiesz (with simple cross-fitting and multitasking)

|        | Bias   | RMSE  | Cov.  |
|--------|--------|-------|-------|
| DR     | −0.131 | 0.377 | 0.909 |
| Direct | 0.094  | 0.304 | 0.046 |
| IPS    | −0.161 | 0.443 | 0.847 |
| TMLE-DR| −0.082 | 0.327 | 0.953 |
| **Benchmark:** | | | |
| RF Plug-in | 0.055 | 0.327 | 0.912 |

Figure 2 shows the distribution of estimates under the most complex design for RieszNet and Forest-Net (simple cross-fitting and multitasking). The distribution is approximately normal and properly centered around the true value, with very small bias for the doubly-robust estimators.

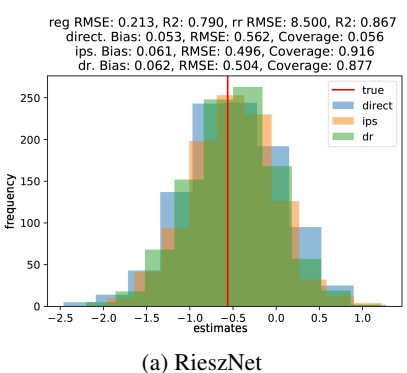
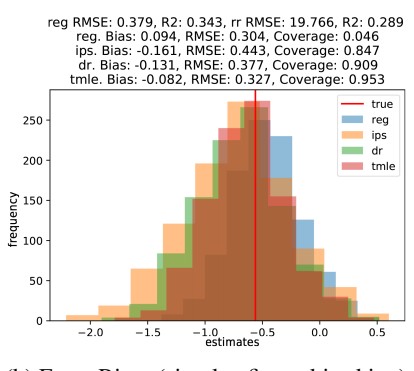

(a) RieszNet     (b) ForestRiesz (simple xfit, multitasking)

Figure 2: RieszNet and ForestRiesz: Regression and Riesz representer $R^2$ and RMSE, bias, RMSE, coverage and distribution of estimates over 1000 semi-synthetic datasets based on the BHP gasoline price data (10 different random seeds). The DGP is based on a complex regression function with linear and non-linear confounders.

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

## A  APPENDIX

### A.1  RIESZNET ARCHITECTURE AND TRAINING DETAILS

As described in Section 3, the architecture of RieszNet consists of $k$ common hidden layers that are used to learn both the RR and the regression function, and $d - k$ additional hidden layers to learn the regression function. In our simulations, we choose $k = 3$ with a width of 200 and ELU activation function for the common hidden layers, and $d - k = 2$ with a width of 100 and also ELU activation function for the regression hidden layers.

We split our dataset in a training fold and a test fold (20% and 80% of the sample respectively). Following Shi et al. (2019), we train our network in two steps: (i) a fast training step, (ii) a fine-tuning step. In the fast training step, we use a learning rate of $10^{-4}$, with early stopping after 2 epochs if the test error is smaller than $10^{-4}$, and with a maximum of 100 training epochs. In the fine tuning step, we use a learning rate of $10^{-5}$, with the same early stopping rule after 40 epochs, and with a maximum of 600 training epochs.

We use L2 regularization throughout, with a penalty of $10^{-3}$, a weight $\lambda_1 = 0.1$ on the RRLoss and $\lambda_2 = 1$ on the targeted regularization loss (as defined in Equation (7)), and the Adam optimizer.

### A.2  DESIGNS FOR THE BHP EXPERIMENT

For the average derivative experiment based on BHP data, we generate the outcome variable $y = f(T, X) + \epsilon$ with six different choices of $f$:

1. Simple $f$:
$$f(T, X) = -0.6T$$

2. Simple $f$ with linear confounders:
$$f(T, X) = -0.6T + X_{1:21} \cdot b$$

3. Simple $f$ with linear and non-linear confounders:
$$f(T, X) = -0.6T + X_{1:21} \cdot b + \text{NL}(X)$$

4. Complex $f$:
$$f(T, X) = -\frac{1}{6} \left( \frac{X_1^2}{10} + 0.5 \right) T^3$$

5. Complex $f$ with linear confounders:
$$f(T, X) = -\frac{1}{6} \left( \frac{X_1^2}{10} + X_{1:9} \cdot c + 0.5 \right) T^3 + X_{1:21} \cdot b$$

6. Complex $f$ with linear and non-linear confounders:
$$f(T, X) = -\frac{1}{6} \left( \frac{X_1^2}{10} + X_{1:9} \cdot c + 0.5 \right) T^3 + X_{1:21} \cdot b + \text{NL}(X)$$

where $\text{NL}(X) = 1.5\sigma(10X_6) + 1.5\sigma(10X_8)$ for the sigmoid function $\sigma(t) = 1/(1 + e^{-t})$, and where the coefficients $b \sim$ iid $\mathcal{U}[-0.5, 0.5]$ and $c \sim$ iid $\mathcal{U}[-0.2, 0.2]$ are drawn once per design at the beginning of the simulations (we try 10 different random seeds).

### A.3 FULL SET OF RESULTS FOR THE BHP EXPERIMENT

Table A1: RieszNet: Regression and Riesz representer $R^2$, bias, RMSE and coverage over 1000 semi-synthetic datasets based on the BHP gasoline price data (10 different random seeds).

| | | Direct | | | IPS | | | DR | | |
|---|---|---|---|---|---|---|---|---|---|---|
| reg $R^2$ | rr $R^2$ | Bias | RMSE | Cov. | Bias | RMSE | Cov. | Bias | RMSE | Cov. |
| **1. Simple $f$** | | | | | | | | | | |
| 0.890 | 0.871 | 0.008 | 0.047 | 0.063 | 0.022 | 0.044 | 0.939 | 0.009 | 0.041 | 0.926 |
| **2. Simple $f$ with linear confound.** | | | | | | | | | | |
| 0.825 | 0.865 | 0.046 | 0.554 | 0.047 | 0.083 | 0.494 | 0.918 | 0.065 | 0.500 | 0.872 |
| **3. Simple $f$ with linear and non-linear confound.** | | | | | | | | | | |
| 0.789 | 0.866 | 0.044 | 0.563 | 0.047 | 0.070 | 0.499 | 0.914 | 0.058 | 0.506 | 0.878 |
| **4. Complex $f$** | | | | | | | | | | |
| 0.852 | 0.873 | 0.021 | 0.064 | 0.107 | 0.020 | 0.053 | 0.957 | 0.021 | 0.056 | 0.920 |
| **5. Complex $f$ with linear confound.** | | | | | | | | | | |
| 0.826 | 0.864 | 0.057 | 0.548 | 0.051 | 0.074 | 0.491 | 0.924 | 0.072 | 0.500 | 0.878 |
| **6. Complex $f$ with linear and non-linear confound.** | | | | | | | | | | |
| 0.790 | 0.867 | 0.053 | 0.562 | 0.056 | 0.061 | 0.496 | 0.916 | 0.062 | 0.504 | 0.877 |

Table A2: ForestRiesz: Regression and Riesz representer $R^2$, bias, RMSE and coverage over 1000 semi-synthetic datasets based on the BHP gasoline price data (10 different random seeds).

| x-fit | multit. | reg $R^2$ | rr $R^2$ | Direct Bias | RMSE | Cov. | IPS Bias | RMSE | Cov. | DR Bias | RMSE | Cov. | TMLE-DR Bias | RMSE | Cov. |
|---|---|---|---|---|---|---|---|---|---|---|---|---|---|---|---|
| **1. Simple $f$** | | | | | | | | | | | | | | | |
| 0 | Yes | 0.952 | 0.491 | 0.133 | 0.134 | 0.000 | 0.148 | 0.149 | 0.000 | 0.043 | 0.050 | 0.424 | 0.003 | 0.028 | 0.844 |
| 0 | No | 0.960 | 0.491 | 0.112 | 0.113 | 0.000 | 0.148 | 0.149 | 0.000 | 0.031 | 0.039 | 0.701 | −0.005 | 0.028 | 0.844 |
| 1 | Yes | 0.931 | 0.290 | 0.136 | 0.138 | 0.000 | −0.096 | 0.102 | 0.222 | −0.026 | 0.037 | 0.892 | 0.009 | 0.025 | 0.962 |
| 1 | No | 0.945 | 0.290 | 0.114 | 0.116 | 0.000 | −0.096 | 0.102 | 0.222 | −0.029 | 0.039 | 0.848 | 0.002 | 0.024 | 0.962 |
| 2 | No | 0.919 | 0.231 | 0.147 | 0.149 | 0.000 | −0.091 | 0.099 | 0.328 | −0.022 | 0.038 | 0.894 | 0.025 | 0.036 | 0.864 |
| **2. Simple $f$ with linear confound.** | | | | | | | | | | | | | | | |
| 0 | Yes | 0.342 | 0.490 | 0.170 | 0.289 | 0.051 | 0.182 | 0.292 | 0.876 | 0.078 | 0.297 | 0.856 | 0.037 | 0.317 | 0.835 |
| 0 | No | 0.711 | 0.491 | 0.194 | 0.305 | 0.047 | 0.183 | 0.293 | 0.875 | 0.073 | 0.274 | 0.881 | 0.019 | 0.298 | 0.848 |
| 1 | Yes | 0.355 | 0.289 | 0.157 | 0.323 | 0.040 | 0.044 | 0.409 | 0.883 | 0.082 | 0.353 | 0.932 | 0.099 | 0.320 | 0.952 |
| 1 | No | 0.665 | 0.290 | 0.188 | 0.306 | 0.047 | 0.044 | 0.409 | 0.882 | 0.074 | 0.340 | 0.932 | 0.099 | 0.324 | 0.942 |
| 2 | No | 0.510 | 0.231 | 0.129 | 0.326 | 0.042 | 0.024 | 0.407 | 0.895 | 0.065 | 0.435 | 0.855 | 0.082 | 0.331 | 0.948 |
| **3. Simple $f$ with linear and non-linear confound.** | | | | | | | | | | | | | | | |
| 0 | Yes | 0.330 | 0.490 | 0.154 | 0.283 | 0.051 | 0.163 | 0.284 | 0.888 | 0.057 | 0.296 | 0.862 | 0.014 | 0.320 | 0.842 |
| 0 | No | 0.684 | 0.491 | 0.168 | 0.295 | 0.055 | 0.164 | 0.284 | 0.888 | 0.048 | 0.274 | 0.884 | −0.005 | 0.302 | 0.850 |
| 1 | Yes | 0.342 | 0.289 | 0.136 | 0.317 | 0.044 | −0.004 | 0.413 | 0.878 | 0.040 | 0.353 | 0.934 | 0.061 | 0.317 | 0.955 |
| 1 | No | 0.639 | 0.290 | 0.159 | 0.295 | 0.051 | −0.004 | 0.413 | 0.880 | 0.045 | 0.342 | 0.939 | 0.069 | 0.323 | 0.954 |
| 2 | No | 0.488 | 0.231 | 0.105 | 0.322 | 0.059 | −0.021 | 0.415 | 0.898 | 0.016 | 0.433 | 0.861 | 0.041 | 0.327 | 0.961 |
| **4. Complex $f$** | | | | | | | | | | | | | | | |
| 0 | Yes | 0.866 | 0.491 | 0.076 | 0.081 | 0.010 | 0.082 | 0.086 | 0.299 | −0.026 | 0.042 | 0.813 | −0.072 | 0.081 | 0.299 |
| 0 | No | 0.866 | 0.491 | 0.049 | 0.058 | 0.020 | 0.082 | 0.086 | 0.299 | −0.039 | 0.051 | 0.691 | −0.078 | 0.086 | 0.246 |
| 1 | Yes | 0.734 | 0.290 | 0.097 | 0.101 | 0.000 | −0.259 | 0.264 | 0.000 | −0.201 | 0.206 | 0.000 | −0.137 | 0.141 | 0.038 |
| 1 | No | 0.735 | 0.290 | 0.072 | 0.081 | 0.000 | −0.259 | 0.264 | 0.000 | −0.208 | 0.212 | 0.000 | −0.147 | 0.151 | 0.018 |
| 2 | No | 0.730 | 0.231 | 0.113 | 0.118 | 0.000 | −0.272 | 0.278 | 0.000 | −0.200 | 0.205 | 0.008 | −0.112 | 0.117 | 0.206 |
| **5. Complex $f$ with linear confound.** | | | | | | | | | | | | | | | |
| 0 | Yes | 0.343 | 0.490 | 0.111 | 0.262 | 0.060 | 0.114 | 0.258 | 0.927 | 0.008 | 0.289 | 0.875 | −0.037 | 0.319 | 0.835 |
| 0 | No | 0.710 | 0.491 | 0.096 | 0.263 | 0.060 | 0.114 | 0.258 | 0.926 | −0.009 | 0.267 | 0.895 | −0.056 | 0.304 | 0.838 |
| 1 | Yes | 0.356 | 0.289 | 0.115 | 0.308 | 0.049 | −0.113 | 0.426 | 0.856 | −0.089 | 0.361 | 0.916 | −0.044 | 0.316 | 0.959 |
| 1 | No | 0.663 | 0.290 | 0.098 | 0.268 | 0.064 | −0.113 | 0.426 | 0.855 | −0.106 | 0.351 | 0.931 | −0.061 | 0.320 | 0.951 |
| 2 | No | 0.509 | 0.231 | 0.094 | 0.315 | 0.056 | −0.150 | 0.446 | 0.860 | −0.109 | 0.441 | 0.845 | −0.052 | 0.327 | 0.952 |
| **6. Complex $f$ with linear and non-linear confound.** | | | | | | | | | | | | | | | |
| 0 | Yes | 0.332 | 0.490 | 0.095 | 0.259 | 0.064 | 0.095 | 0.253 | 0.936 | −0.012 | 0.293 | 0.884 | −0.060 | 0.326 | 0.835 |
| 0 | No | 0.683 | 0.491 | 0.070 | 0.260 | 0.053 | 0.095 | 0.253 | 0.934 | −0.033 | 0.274 | 0.894 | −0.079 | 0.314 | 0.827 |
| 1 | Yes | 0.343 | 0.289 | 0.094 | 0.304 | 0.046 | −0.161 | 0.443 | 0.847 | −0.131 | 0.377 | 0.909 | −0.082 | 0.327 | 0.953 |
| 1 | No | 0.637 | 0.290 | 0.069 | 0.264 | 0.056 | −0.160 | 0.443 | 0.846 | −0.135 | 0.365 | 0.911 | −0.091 | 0.331 | 0.945 |
| 2 | No | 0.488 | 0.231 | 0.070 | 0.313 | 0.061 | −0.196 | 0.468 | 0.849 | −0.158 | 0.454 | 0.831 | −0.094 | 0.338 | 0.950 |

Table A3: RF Plug-in Benchmark: Regression and Riesz representer $R^2$, bias, RMSE and coverage over 1000 semi-synthetic datasets based on the BHP gasoline price data (10 different random seeds).

| reg $R^2$ | rr $R^2$ | RF Plug-in | | | reg $R^2$ | rr $R^2$ | RF Plug-in | | |
| | | Bias | RMSE | Cov. | | | Bias | RMSE | Cov. |
|---|---|---|---|---|---|---|---|---|---|
| **1. Simple $f$** | | | | | **4. Complex $f$** | | | | |
| 0.960 | 0.491 | 0.043 | 0.051 | 0.535 | 0.866 | 0.491 | 0.030 | 0.048 | 0.791 |
| **2. Simple $f$ with linear confound.** | | | | | **5. Complex $f$ with linear confound.** | | | | |
| 0.711 | 0.491 | 0.090 | 0.328 | 0.912 | 0.710 | 0.491 | 0.065 | 0.323 | 0.914 |
| **3. Simple $f$ with linear and non-linear confound.** | | | | | **6. Complex $f$ with linear and non-linear confound.** | | | | |
| 0.684 | 0.491 | 0.080 | 0.332 | 0.915 | 0.683 | 0.491 | 0.055 | 0.327 | 0.912 |

