# OpenReview forum: "RieszNet and ForestRiesz: Automatic Debiased Machine Learning with Neural Nets and Random Forests"
_ICLR.cc/2022/Conference — ICLR 2022 Submitted_

### Official Review · Reviewer_WPay · 2021-11-02

**Correctness:** 4
**Technical Novelty And Significance:** 2
**Empirical Novelty And Significance:** 2
**Recommendation:** 5
**Confidence:** 4

**Main Review:**

Main review:
- The proposed method is very innovative and important in the field. This paper will be of great value to the field.
- On the other hand, it is difficult to evaluate the novelty of the paper because many of the results rely on existing studies, such as Chernozhukov et al. (2021).
- This paper may be more of a summary of the existing studies.
- If the contribution of this paper is the proposal of the methods, then the experiments to show the validity of the methods will be important as the contribution of this paper.
- However, the experiments are very simple, and it is not clear whether they are superior to other methods.
- Even though the performance depends on the architecture and hyperparameters of the model, there is a little description of them.
- Besides, although the proposed method is persuasive, it seems to me that the paper is not well organized. There were several points that I couldn't catch up on. For instance,
   - What are the definitions of $Y$, $X$, $Z$, $W$, and $T$? We can guess the meanings, but the rigorous definitions are needed.
   - In Example 3,  $X=(T, X)$ should be $Z=(T, X)$?
   - Theorem 1 is cited from Chernozhukov et al. (2021), but there is no description about which theorem is used from Chernozhukov et al. (2021).
   - In the first set in Equation (3), the authors might have forgotten to add $\gamma \in [0,1]$.
   - In page 4, what are the definitions of $\beta$, $w_{1:k}$, $w_{1:d}$, $d$, and $k$? I can guess about them, but the clear definitions are needed.
   - Equation (5) and (6) are correct? The LHS of these equations are not functions of $w$ because they are minimized over $w$. I think that $min$ operator should be removed.
   - In the RHS of Equation (5), $\theta$ is $\beta$?
   - What are the definitions of $f_2$ and $f$?

Questions:
1. Can we apply the proposed method to estimate the conditional (potentially heterogeneous) average treatment effect?
2. Can we check that the conditions of Theorem 1 hold? In particular, the multi-task neural network architecture can satisfy them?
3. The estimator has $\sqrt{n}$ consistency? I think that we can derive the convergence rate from Theorem 1, which states the convergence rate of $\|\hat{\alpha} - \alpha_0\|^2_2$, but it is not clear, at least for me.

Minor points:
1. Figures 2 and 3 are a little difficult to see because the histograms overlap. How about using boxplots?

**Summary Of The Paper:**

This paper shows novel methods based on the recent findings on the Riesz representation in econometrics and machine learning, such as Chernozhukov et al. (2021). While the proposed applications make sense and are persuasive, the contributions of this paper are a bit limited. Although the authors support the soundness of the proposed algorithms in experiments, the experiments are a bit simple and may not be sufficient for the justification.

**Summary Of The Review:**

I find it difficult to evaluate the contribution of this paper because it is not clear what the contribution of this paper is. The method is novel, but the theory relies on existing different work, so I think the evaluation of the method highly relies on the novelty of experiments. However, the experiments are very simple and not enough to claim the usefulness of the method. If the methodological and theoretical contributions are significant, the paper should be highly evaluated even if the experiments are simple. However, in this case, because this paper has less novelties in  the theoretical part, the experiments are important. In summary, the method in this paper is very novel and interesting, but the experiments to support it are lacking. Besides, there are several issues in writing even though we can guess what the authors insist. This is the reason why I vote for weak rejection.

---

> ### Author Response · Authors · 2021-11-23
> **Reply to Reviewer WPay**
>
> Thank you so much for your helpful comments and suggestions. We have done our best to explain the notation and fix the typos in this new version of the paper.
>
> We have also tried to explain more carefully what is new and what was previously known. The debiasing technique is not new: e.g. Chernozhukov et al. (2018a) already show how to use the Riesz representer to construct debiased, consistent and asymptotically normal estimates of the parameters of interest. What is novel is the implementation using neural networks and random forests of an Auto-DML technique (the RR is estimated directly, rather than by plugin, which makes it suitable even for functionals whose RR does not have a known form). This complements the work of Chernozhukov et al. (2021), which provides theoretical guarantees for generic machine learners. This paper focuses on (a) the practical implementation through RieszNet and ForestRiesz, and (b) evaluating these in experiments and showing improved performance against plugin methods. The implementation and empirical evaluation are completely new.
>
> We tried to evaluate our experiments in settings of empirical relevance, that is why we considered the ATE and the average derivative as two applications that many readers considering using these methods in applied work may be interested in. At the same time, both the theory and the implementation of RieszNet and ForestRiesz are more general than that. We think that, besides evaluating the performance of our methods, the paper adds value to the literature in introducing two ready-to-use implementations of the Auto-DML generic framework of Chernozhukov et al. (2021).
>
> To your specific comments on notation:
> 1. We have added definitions in words of what we have in mind for most applications. Each example also contains some more details.
> 2. Yes, thank you for spotting this typo, it is now fixed.
> 3. Sorry it was not clear, this is also Theorem 1 in Chernozhukov et al. (2021).
> 4. Yes, thank you for spotting this typo, it is now fixed.
> 5. We have added the corresponding definitions.
> 6. We have restructured these equations to make it more clear. We have removed the $\min$ operator, since, in the end, what we minimize is equation (7).
> 7. Yes, thank you for spotting this typo, it is now fixed.
> 8. These are arbitrary deep neural networks ($f$ is now renamed $f_1$). We have expanded the explanations in the main text, and added an appendix detailing the exact architecture of these in our experiments.
>
> Questions:
> 1. Conditional treatment effects do not directly fall in the framework of this paper, although work guided by similar principles exist to estimate CATEs (e.g. Semenova & Chernozhukov, 2020)
> 2. That is a very important question, although it is not the focus of this paper. The rates for neural net estimators under different assumptions are discussed in a related theory paper  by Chernozhukov et al. (2021).
> 3. The DR estimator of $\widehat{\theta}$ is $\sqrt{n}$-consistent and asymptotically normal if the product of the estimation errors for $\alpha_0$ and $g_0$ vanishes fast enough. In particular, for settings where the regression function can be estimated very well, the requirements on the RR rate will be less strict, and vice versa. These details are also discussed in a related theory paper by Chernozhukov et al. (2021).
>
> Minor points: We have kept them as histograms, since what we wanted to convey is the approximately normal distribution of the estimates. The most important information (bias, RMSE, coverage) is in the text above the figure.

---

### Official Review · Reviewer_DMWH · 2021-11-02

**Correctness:** 3
**Technical Novelty And Significance:** 2
**Empirical Novelty And Significance:** 3
**Recommendation:** 6
**Confidence:** 4

**Main Review:**

I like the premise of the paper. I think it is an important step towards making the estimation of policy-relevant causal estimands more accessible to practitioners working in data-rich environments in which nuisance functions are expected to be complicated. Although conceptually and theoretically most of the content of the paper is already known / described elsewhere, its novelty lies in combining multiple ingredients to propose practical methods and provide a concrete implementation.

I have some suggestions for improving the paper:

1) Compared to other papers in this line of work, it appears to me that the scope of the paper is relevant to a broader audience and especially so for applied data scientists. Yet,
the paper feels rushed (more on this below), e.g., with a lot of notation remaining undefined. The consequence is that the paper is not self-contained and would be very hard to understand, even for somebody with background knowledge in causality (who is not aware of the latest theoretical developments). Exposition is very important for a paper that could reach a broad audience!

2) I felt the description of the methods is a bit too short and probably one would not be able to reproduce the results based on the text description. This is particularly the case for the ForestRiesz procedure which seems to me to be less "automated" than promised.

3) There are many ingredients to the proposed procedures (beyond what I described above) and from the experiments it is not clear which ingredient is relevant (or if all ingredients are relevant). Ablation studies would be very helpful here for the reader.

Below I summarize these three points in more detail:

# Exposition

1) The introduction is very short and one immediately is faced with a formula, the components of which are never defined. It would be very helpful to at least define potential outcome notation early. $W$ is also never defined.

2) I think it would be very helpful to a reader to discuss the assumptions of Theorem 1 in more detail. What is the take-away for a practitioner? What should they be careful about?

3) After Lemma 1, please make sure to cite Rosenbaum and Rubin (1983). I think it would also be helpful to make a remark of how Lemma 1 is used. The "naive" (and perhaps traditional) approach would be to just nonparametrically regress on $\hat{\alpha_0}$. But this requires very small estimation errors for $\alpha_0$. Instead the innovation is the use of e.g., the last layer of the neural network, which may still be capturing all relevant conditioning information!

4) The first paragraph of page 4 talks about cross-fitting and double cross-fitting. What is actually being used in the implementation? For example if double cross-fitting is being used, then I think the reference to that could be removed (the audience for whom this point is relevant would likely already be aware of this point and practitioners would use the "automatic" software).

5) In Section 3, I think that having a figure on the architecture (similar to Fig. 1 of Shi et al.) could be very helpful for a reader.

6) Page 8, Riesz representer with Gaussian errors: It could be worthwhile to point out here that this is the case by Stein's identity (and point a reader e.g. to a supplement that explains this).


# Details for procedures

1) I think it is very difficult to follow what ForestRiesz is doing, e.g., it is not explained what is being computed in the leaves.

2) Also the caveats and description in the first paragraph make the whole procedure seem less "automatic" and the implementation appears to need substantial additional modifications to account for treatments. This makes the procedure less appearling than RieszNet.

3) Please provide more details for the simulations and implementation of procedures, e.g., how many folds are being used? Where is IPS defined? Could there be more details for architectures and tuning parameters chosen? What is the "RF Plug-in" benchmark? Is it the default recommended method in the GRF package?

# Ablation studies

For example for the ATE simulations I would be interested in ablation studies showcasing the following aspects:

1) What is the impact of the regularization term for RieszNet?

2) What is the impact of cross-fitting? Shi et al. (surprisingly to me) claim that cross-fitting does not help for their DragonNet architecture.


# Some potential Typos / minor points

* Page 2, "integration by parts gives" -> Is a negative sign missing?
* Page 2, Linearity after the first eqn. of Section 2?
* Page 3, after eqn. (3), "square" on the RHS?
* Page 4, before eqn. (6), $(k+1):d$ instead of $k:d$?
* Check formatting of references, e.g., Grathwohl et al. after example 4 should presumably be in a parenthesis?







**Summary Of The Paper:**

The authors study the nonparametric estimation of an important class of (causal) estimands that includes the average treatment effect (ATE) in experiments and observational studies under unconfoundedness. The main innovation is the "automatic" nature of the procedures, one based on deep learning and one on random forests. While previous work has developed hand-tailored constructions for the ATE (a traditional problem in statistics), the authors show that existing constructions can be generalized considerably based on recent advances in semi- and nonparametric statistics based on Riesz representers. The key challenge this work addresses is the fact that the Riesz representer is typically unknown.

The main proposed method based on deep learning, RieszNet, may be seen as a generalization of the DragonNet procedure by Shi et al. (2019) from the ATE to more general estimands. Even in the case of the ATE, DragonNet and RieszNet are not the same and RieszNet outperforms DragonNet in simulations: RieszNet directly targets the inverse propensities, while DragonNet estimates the propensities and then plugs in their inverse.

**Summary Of The Review:**

This paper provides a solid contribution towards the automatic estimation of policy-relevant causal estimands. I think this work should be accepted. However the paper could be substantially improved by improving exposition and making the paper accessible to a wider audience.

---

> ### Author Response · Authors · 2021-11-23
> **Reply to Reviewer DMWH**
>
> Thank you so much for your helpful comments and suggestions. We have done our best to explain the notation and fix the typos in this new version of the paper.
>
> We do believe that our paper has relevance for a broader audience. That is why we selected the ATE and average derivative experiments, as they are something many applied researchers are interested in. We have tried to define the notation more carefully and to fill in some missing details, towards which your suggestions and the other reviewers’ have been a great help. We also provide replication code, which can be easily adapted to other settings.
>
> To your specific comments on the exposition:
> 1. We have tried not to use potential outcome notation in the paper for simplicity. We do mention potential outcomes tangentially in the ATE example, with a reference to Rosenbaum and Rubin (1983) for the interested reader. It is true that $W = (Y,Z)$ was never defined. We have added that, as well as an explanation in words of what $Y$ and $Z = (T,X)$ will be in typical applications. Each example also contains some more details.
> 2. The first assumption is mostly technical, and it allows us to obtain a rate of convergence for $\widehat{\alpha}$ based on the critical radius of the space of approximating functions. The mean-squared continuity property of $m$ is satisfied by our examples and most functionals of interest. For a more technical discussion of the assumption, we refer the reader to Chernozhukov et al. (2021).
> 3. Thank you, we have added the citation. We explain what the Lemma means in words as the last sentence of the Lemma, and we discuss how exactly we apply it to RieszNet right after equation (5). This observation is not fully new, since Shi et al. (2019) base their dragonnet architecture on a similar principle.
> 4. Most experiments use no crossfitting. In the average derivative experiment, we try ForestRiesz with different types of cross-fitting (simple and double). For RieszNet, we found that cross-fitting did not improve results greatly compared to the increase in computation time.
> 5. We have added some details on the architecture in an Appendix, hopefully those will be useful in making the architecture clearer to the reader.
> 6. Thanks, we have added that remark.
>
> Details for procedures:
> 1. In its multitasking version, ForestRiesz uses a GRF based on the Riesz loss of equation (8) and the squared loss for the regression (which has F.O.C. given by the inline expression in the last paragraph of section 4).
> 2. What we have in mind when we say that our methods are “automatic” is that they do not require knowledge of the specific form of $\alpha_0$, but only being able to evaluate $m$. In particular, in the binary treatment case $T \in \lbrace 0,1\rbrace$ notice that writing $\alpha(T,X) = T \alpha(1, X) + (1 - T) \alpha(0,X)$ is fully without loss of generality.
> 3. We have added more details. IPS is now defined at the start of Section 5. The “RF Plug-in” is defined within 5.1. and 5.2. It is not the default in the GRF package. It uses the known explicit form of the RR in our examples to estimate it. In 5.1., it requires estimating the propensity score; in 5.2., it requires the conditional mean and variance of $T$ given $X$; all these quantities are estimated with random forests.
>
> Ablation studies: we agree that these questions are very interesting and important. Due to limitations in space and time, we have not added these to the current version of the paper, but it is definitely something we want to investigate further in the future. We also provide replication files, so that the reader can experiment at will with hyperparameters such as regularization.
>
> Typos: Thank you for spotting them, they are now fixed.

---

### Official Review · Reviewer_S7Am · 2021-11-08

**Correctness:** 3
**Technical Novelty And Significance:** 3
**Empirical Novelty And Significance:** 3
**Recommendation:** 6
**Confidence:** 3

**Main Review:**


Strengths
- The proposed methodology that the estimation problem of the moment function is converted to the problem of learning the Riesz representer seems to be novel. To learn the Riesz representer, the authors introduce a loss function according to a theoretical result of recent work.
- The author proposed two methods to learn Riesz representer, including a Neural Network method (RieszNet) and a random forest method (ForestRiesz). The experimental evaluation seems to be very strong and can achieve state-of-the-art performance in estimating the average treatment effect.

Weakness
- The notations in this paper are very confusing. For example, $W$, $W_i$, $w_{1:k}$, $w_{1:d}$, $w_{k:d}$ are never defined in this paper. The notation $w$ is abused, referring to a function in Examples 2 and 4, but refering some certain values with a subscript such as $w_{1:k}$.
- The contributions of this paper is unclear. It would be better to clarify which parts are the contributions of this paper and which parts are results from previous works. Some parts of this paper are the conclusions of previous works, while some parts of this paper are methodologies and discussions of comparing methods, but these parts are all mixed together.
- The detail steps of the proposed RieszNet and ForestRiesz should be clarified. For RieszNet, if the network architecture and training details of RieszNet are similar to Shi et al. (2019)? What $w_{1:k}$, $w_{1:d}$, $w_{k:d}$ are referred to, and after the minimization of equation (7) what should we do? For ForestRiesz, if readers are not familiar with Athey et al. (2019), it would be very hard for readers to follow the idea of ForestRiesz.
- Although the paper aims to solve a general problem of estimating the average value of a moment function, the focus of this paper is still causal inference (example 1 average treatment effect). In the experimental part, the paper only evaluates the average treatment effect and average marginal effects, which are both problems in causal inference.




**Summary Of The Paper:**

The authors address the problem of estimating the average value of a moment function that depends on an unknown regression function, which is commonly used in causal inference.
By the Riesz representation theorem, the authors design a loss function where the Riesz representer turns out to be the minimizer of the proposed loss.
The authors propose both a Neural Network method (RieszNet) and a random forest method (ForestRiesz), i.e. a multi-tasking Neural Network method where the loss function is a combination of the Riesz representer and regression loss, and a random forest method that leans representation of both the regression function and the Riesz function.
The authors conduct experiments of estimating the average treatment effect and average marginal effects, and show that the proposed RieszNet and ForestRiesz beat the state-of-the-art methods.




**Summary Of The Review:**

The idea of this paper is original, and both RieszNet and ForestRiesz show strong experimental evaluations. But the paper needs to be well-organized. Therefore, I would recommend acceptance of this paper if the paper was well-written.

---

> ### Author Response · Authors · 2021-11-23
> **Reply to Reviewer S7Am**
>
> Thank you so much for your helpful comments and suggestions. We have done our best to explain the notation and fix the typos in this new version of the paper.
>
> We have also tried to state more clearly what our contribution is. In terms of the structure of the paper, sections 1 and 2 provide background, based on existing results (including the related theory paper Chernozhukov et al., 2021). Sections 3 and 4 introduce the RieszNet and ForestRiesz estimators, which are new. The purpose of Section 5 is to evaluate the performance of these estimators in two experimental settings and to compare them to different benchmarks.
>
> To your other comments:
> * We use $W = (Y,Z)$ to refer to all data, which is now defined at the beginning. We did abuse the $w$ notation, which should be fixed now. The policy function in examples 2 and 4 has been updated to $\pi(X)$. We keep using lowercase $w$ with subindices to refer to the weights of the neural network in RieszNet, but that is now more clearly explained in Section 3.
> * We have added an appendix with details on the architecture and tuning parameters that we use for RieszNet in our experiments. Note that both RieszNet and ForestRiesz are estimators of $\alpha_0$ and $g_0$: once we have those estimators (minimizing eq. (7)) we plug them in into a score that estimates $\theta$. Our preferred score is the DR in equation (1), but in the experiments we also consider the Direct and IPS scores, which are now defined at the beginning of Section 5.
> * We tried to evaluate our experiments in settings of empirical relevance, that is why we considered the ATE and the average derivative as two applications that many readers considering using these methods in applied work may be interested in. That being said, both the theory and the implementation of RieszNet and ForestRiesz are more general than that, and can be used for other settings where we wish to estimate the average value of a functional of an unknown regression function.

---

### Official Review · Reviewer_47Pi · 2021-11-08

**Correctness:** 3
**Technical Novelty And Significance:** 3
**Empirical Novelty And Significance:** 3
**Recommendation:** 8
**Confidence:** 2

**Main Review:**

**Strengths:** The method presented in the paper is applicable to a broad class of problems. The main idea presented in Lemma 1 is easy to understand and is applicable to other machine learning methods as well. The presented methods perform well in experiments, exceeding the performance of two alternative methods, although this comparison is only performed on synthetic variations of one real-world data set. Code is provided.

**Weaknesses:** As someone who is not familiar with the type of problem discussed in the problem, I found some parts of the paper confusing (see below). Some explanations might be missing altogether unless I missed them. I think the paper could benefit from investing more time in the writing, including the notation. Additionally, in my opinion it could be stated more clearly what is and what is not novel about the presented methods (see below). Moreover, it appears to me that the paper hints at some theoretical guarantees in the beginning but then does not really discuss what theoretical guarantees can be given for the presented methods.

**Specific comments / questions:**
1. In the first eq. on page 1: The notation is confusing to me. It is not clear which variables the expectation is taken over. I assume the expectation is over $Z$. Also, it is not clear that $m$ even depends on $Z$. Moreover, in the following examples it is not clear whether $W = w$ or $W = w(Z)$ or something else. It would probably more clear to write something like $\theta_0 = \mathbb{E}_Z [m(Z; w, g_0)]$, or is it even $\mathbb{E}_Z [m(Z; w(Z), g_0)]$?
2. It seems to me that $w$ is always fixed throughout the paper except for the second function space definition in Eq. (3). Hence, it is not clear to me why it would not just be absorbed into the definition of $m$, such that we can write $m(g_0)$. Is the explicit dependence on $w$ in Eq. (3) needed to show that the resulting constant $C$ is uniform over different choices of $w$?
3. The semicolon notation $m(W; g_0)$ is not used consistently throughout the paper.
4. Sec 2: "This framework gives rise to a score": Which framework? Is it new or from the literature? In general, the steps here are not cited and the formulation does not make it clear whether they are novel.
5. Eq (1): What does the index i mean in this context? Is it fixed or summed over? And why does $W$ even have an index? Is it because $W_i = w(X_i)$ ($W$ is not explicitly defined anywhere)? Explaining this might also make the paragraph below more understandable.
6. Theorem 1: Why does $\delta_n$ depend on $n$ even though the function spaces do not depend on $n$? Why is the assumption on all $f$ in the spaces of Equation (3) not formulated directly after the definition of the function spaces? Why would it be reasonable to assume that all functions satisfy $\|f\|_\infty \leq 1$ in the considered settings?
7. Theorem 1: In the mean-squared continuity property equation, is the square root intentionally only taken on the left-hand side and not on the right-hand side? In this case, I would not call it "mean-squared continuity" because then it is not the same as in the beginning of Section 2.
8. Sec 4: I don't understand the first paragraph. "One could potentially minimize the Riesz loss ..." - Why would an empty node change the Riesz loss? In general, the Riesz loss is formulated in terms of $Z = (X, T)$, making no distinction between $X$ and $T$. Why would there arise problems in $T$ that do not arise in $X$? Also, in the statement "Such discontinuous in $T$ function spaces will typically not satisfy the mean-squared continuity property", you could refer to the corresponding equation in Theorem 1, or even explain why this is the case.
9. Eq. (9): What does conditioning on $X = x$ mean for the empirical expectation $\mathbb{E}_n$? Is it only allowed for $x$ from the training set?
10. Figure 1: It seems (at least by text search) that the abbreviation IPS is never explained. Ideally, the abbreviations would be explained in the Figure caption. Is there any reason why only MAE is used for evaluation even though the methods are trained on RMSE? If you want to make Figure 1 look more like a typical comparison table in the Deep Learning literature, you could consider the following suggestions:
- Put both random forests and NNs in a single column, name them RieszNet-DR, RieszNet-Direct etc., and cite the sources (e.g. RieszNet-DR (Ours) or Dragonnet (Shi et al., 2019))
- Mark the best-performing method(s) in bold (the differences seem to be at least partially significant)
- Change the LaTeX environment from figure to table

Why is it not possible to have coverage estimates in this setting? What does it mean to "redraw the treatment" to get coverage estimates?

11. Figure 2: I think you should label the axes, and maybe not put experimental results in text form above the plots.
12. Why do you use the auxiliary loss in eq. (7) if you could also use the debiasing term from Eq. (1)? Do I understand it correctly that neither of the debiasing methods are novel?
13. You could mention the experimental evaluation at the end of Section 1, to have a more complete exposition of the contributions of the paper.
14. Sec 2, 3rd paragraph: Why does it debias the naive plug-in estimator? Is it explained anywhere, or in the literature?
15. Could you discuss what theoretical guarantees can be given for (idealized versions of) the presented methods? What can be said about the infimum in Eq. (4)? The asymptotic statement $\delta_n = O(d_n n^{-1/2})$ is not helpful if it is not clear how $d_n$ can behave. Moreover, the constraint $\|f\|_\infty \leq 1$ seems to be restrictive, and even in the best case, the conclusion of the theorem gives what I suppose is meant by parametric $\sqrt{n}$-rates only for $\alpha$ and not for $\theta$.
16. How are the confidence intervals computed?
17. In my opinion Table 2 and Table 1 might be better placed in the Appendix, with perhaps some smaller summary table/figure in the main part of the paper. Also, making the best values bold might aid interpretation of the results. The gained space in the main paper could be used for explaining some other parts in more detail.
18. I did not find any details on the training of the neural networks. Although they are of course contained in the supplementary code, maybe they should be briefly described in the appendix. (Number of layers, optimizer, activation function, etc.)
19. Since the method is not that easy to summarize in the introduction, the paper might benefit from a conclusion section.


**Minor comments:**
1. Example 3: It should be $Z = (T, X)$ instead of $X = (T, X)$
2. Sec 2: "mean-square continuous functional" -> "mean-square integrable continuous linear functional"?
3. Sec 2, second paragraph: maybe it should be $f_0$ instead of $f$
4. Sec 2: Could mention here that $\alpha_0$ is in general unknown and will be estimated.
5. Sec 2: "This framework gives rise to a score": One line uses $\theta$ and the other one $\theta_0$. Is this intentional?
6. Eq (3): should this be a $\mapsto$ instead of a $\to$? Why $a \in \mathcal{A}$ and not $\alpha \in \mathcal{A}$?
7. Sec 2: "focus on objects" -> "focus on problems"?
8. Figure 3: Orthogonality has never been discussed in the paper, or is this standard to report in the literature?
9. Lemma 1: You could place the qed symbol in the same line as the last equation (using \qedhere).
10. The terms "mean-independent" and "parametric rates" were unclear to me. Also, for non-mathematicians, it might be helpful to explain a bit more about Riesz representers.

**Summary Of The Paper:**

This paper considers the problem of estimating an expectation over covariates of some functional of an unknown regression function. The authors propose two estimators, one based on neural networks and one based on random forests. In contrast to many (all?) previous estimators which were derived for specific functionals, these estimators are applicable to general functionals. They employ a (new?) debiasing technique. Moreover, they use a novel multi-task architecture based on the observation that it suffices to estimate the regression function as a function of the Riesz representer. The paper demonstrates improved accuracies compared to prior work and good coverage on semi-synthetic tasks derived from two data sets.

**Summary Of The Review:**

The paper contains interesting new methods which exhibit very good results in a limited experimental evaluation. However, the exposition is lacking in clarity to me. Therefore, I would currently classify the paper as borderline, but I am open to adjust my score based on the authors' response.

**Edit**: Due to the author's response, I raise my score from 6 to 8.

---

> ### Author Response · Authors · 2021-11-23
> **Reply to Reviewer 47Pi**
>
> Thank you so much for your helpful comments and suggestions. We have done our best to explain the notation and fix the typos in this new version of the paper.
>
> We have also tried to state more clearly what our contribution is. As you summed it up, this paper proposes two estimators for causal effects that are defined as the expectation of a functional of an unknown regression function. The debiasing technique is not new: e.g. Chernozhukov et al. (2018a) already show how to use the Riesz representer to construct debiased, consistent and asymptotically normal estimates of the parameters of interest. What is novel is the implementation using neural networks and random forests of an Auto-DML technique (the RR is estimated directly, rather than by plugin, which makes it suitable even for functionals whose RR does not have a known form). This complements the work of Chernozhukov et al. (2021), which provides theoretical guarantees for generic machine learners. This paper focuses on (a) the practical implementation through RieszNet and ForestRiesz, and (b) evaluating these in experiments and showing improved performance against plugin methods.
>
>  To your specific comments and questions, apart from typos:
>
> 1: $W = (Y, Z)$ is a random variable that represents all data. We previously used $w(X)$ to represent a policy in the average policy effect examples. We have now updated that notation to $\pi(X)$.
>
> 2: We write $m(W; g)$ because the functional $m$ can depend on the data directly, as well as through the regression function $g$.
>
> 4: We have rephrased that paragraph. The debiasing term $\alpha(X) (Y - g(X))$ is not new (see e.g. the cited work Chernozhukov et al., 2018a).
>
> 5: We have removed the index $i$ inside the empirical average operator $\mathbb{E}_n$.
>
> 6: The space of functions $\mathcal{A}$ is allowed to grow with $n$ to allow for richer approximations as the sample size increases. To make that clear, we have updated the notation to $\mathcal{A}_n$. The sup-bound of 1 is a technical condition to directly apply Lemma 11 of Foster & Syrgkanis (2020), it could be adapted to a different bound.
>
> 7: That was a typo, we’ve removed the square root. The sense in which mean-square continuity holds should be the same as in Section 2.
>
> 8: The second paragraph of this section tries to clarify this with an example. For the ATE, $m(W; \alpha) = \alpha(1,X) - \alpha(0,X)$, hence at each leaf we want to be able to compute both $\alpha(1,X) $ and $\alpha(0,X)$.
>
> 9: We have updated the notation, since this refers to the population moment. In practice, the same needs to hold at the empirical equivalent of the unconditional moment at the leaf corresponding to $x$.
>
> 10: We have added an explanation of how the direct, IPS and DR methods differ. As for your question about coverage, the MAE experiment considers a fixed treatment $T$ over simulations. Hence, coverage probabilities based on this design ignore the additional uncertainty from the treatment being randomly assigned (conditional on covariates $X$). Thus, in the coverage experiment we “redraw” (i.e. resimulate $T \sim Bernoulli(p(X))$, where the propensity score $p(X)$ is as given in the NPCI R package).
>
> 12: In the RieszNet-DR method we do use eq. (1) as a score to estimate $\theta_0$, with the debiasing term. We add the TMLE loss to (7) as an extra regularization term, following Shi et al. (2019). The debiasing approaches are not novel, what is new is that the RR is estimated “automatically” rather than by plugging-in an estimate for the propensity score.
>
> 14: It is explained briefly in the same paragraph and more extensively in the literature (e.g. Chernozhukov et al. 2018a). The idea is that, once the debiasing term is added, the expectation of the score depends only on the product of the estimation errors $g - g_0$ and $\alpha - \alpha_0$, which vanishes faster than each of those estimation errors separately (at a rate equal to the product of the rates for each of those estimation errors).
>
> 15: The theoretical guarantees are beyond the scope of this work and are discussed more extensively in Chernozhukov et al. (2021), which includes special results for neural networks.
>
> 16: The CIs are computed based on the asymptotic normal approximation $[\widehat{\theta} \mp 1.96 s.e.]$, where the standard error is $\sqrt{\mathbb{E}_n[(\psi - \widehat{\theta})^2]}$ and $\psi$ is the corresponding score used (direct, IPS or DR, as defined now at the beginning of section 5).
>
> 18: We have now added an appendix with this information.
>
> 19: We agree, unfortunately we are already at the page limit. Instead of that, we have tried to expand on our contribution in the introduction.

---

> > ### Author Response · Authors · 2021-11-23
> > **Reply to minor comments**
> >
> > Reply to minor comments:
> >
> > 2:  We have added the qualifier “linear,” since the functionals we consider in this paper are all linear in $g$ (although the theory can be extended to nonlinear functionals, as discussed in Chernozhukov et al. 2021).
> >
> > 5: Yes, the first $\theta$ denotes a generic value, whereas the display equation holds only at the true value of the parameter $\theta_0$
> >
> > 8: This is actually not something we want to focus on, so we have just removed it from the graph.
> >
> > 10: We have rewritten slightly section 2 to explain how the Riesz representer is important for debiasing, we don’t think more details are needed to understand the current paper, which is more applied in nature, but a good reference on the role of the Riesz representer in debiased ML is Chernozhukov et al. (2018a).

---

> > ### Comment · Reviewer_47Pi · 2021-11-29
> > **Response to authors**
> >
> > Thanks for answering most of my points. I read the updated version of the paper. I still have some points and questions, and although the author response period is over, maybe you could try to address some or all of the questions in the final version of the paper.
> >
> > **Some points regarding the updated paper:**
> > - In the introduction, the authors still claim rates (and asymptotic normality) for the resulting estimators. This statement should be weakened unless rates (and asymptotic normality) are proven for RieszNet or ForestRiesz. While some (not very strong) emprical evidence of asymptotic normality is shown, no empirical evidence for rates is shown. In absence of theoretical guarantees, it might be a good idea to perform experiments on increasing amounts of (synthetic) data to hopefully be able to see a difference in rates for Direct/IPS/DR in a log-log plot.
> > - Regarding point 16: This explanation should be in the paper (such that the meaning of $\psi$ and $\hat \theta$ is notationally clear). From the formula for the standard error, it is not clear to me why it could converge to zero with increasing number of samples - should the standard error somehow be divided by $\sqrt{n}$, as is done e.g. on the Wikipedia page on standard error? Since you normally do not use cross-fitting, is the standard error evaluated on samples from the training set? Would the standard error be reliable without strong regularization for RieszNet?
> > - Beginning of Section 4: The first two paragraphs are still very hard to understand for me. The example in the response is more clear: If I understand it correctly, the problem is that the Riesz loss for a fixed value of $W$ does not only depend on a single value of $\alpha$. In the case of the ATE, it depends on $\alpha(1, X)$ and $\alpha(0, X)$. This is still somewhat confusing to me, since a tree that only splits on $X$ could then never improve $\alpha(1, X) - \alpha(0, X)$ and therefore not find sensible splits on $X$, but I suppose that is resolved through the inclusion of the feature map $\phi$. I guess that a specific example like the one in your response would make the beginning of Section 4 more clear. Moreover, if I understand it correctly, some other functionals like $m(W; \alpha) = \alpha(1, X) - \alpha(1, X-1)$ could have the same limitation in $X$ instead of $T$, although they would not be as practically relevant. If that is the case, it should be stated as a limitation of ForestRiesz. (The problem you mention could potentially also be demonstrated with the average marginal effect instead of the ATE, where the RF would have a problem because it does not have nontrivial derivatives in $T$.)
> > - Regarding Table 1 (b), could the authors explain what the difference between Direct and Plug-in is? Is it that the plug-in estimator estimates the propensity score for the Riesz representer, while the Direct method uses the explicit functional?
> > - In Example 1: "If potential outcomes are conditionally independent of treatment $T$ given covariates $X$, then this object is the average treatment effect". Is $Y$ the potential outcome? If $Y$ and $T$ are conditionally independent given $X$, wouldn't that mean that $m(W; g) = 0$ and hence $\theta_0 = 0$?
> > - In Section 5.1, maybe you could explain what the difference between the 1000 semi-synthetic data sets is? Apparently, the treatment is not different - do you redraw the treatment only for coverage because this makes the data sets more realistic in the MAE experiment?
> > - Sec 2.2: In the equation before eq (2), maybe a different hypothesis space should be used (e.g. $\mathcal{A}$ instead of $\mathcal{A}_n$, or even the space of all measurable functions). This would then make more clear that Eq. (2) is an approximation both w.r.t. the hypothesis space and the empirical expectation.
> > - Sec 2.2: $\mathcal{A}_n$ is not consistently used
> > - Sec 2.2: "equation 2" -> "Equation (2)"?
> > - page 3 bottom: maybe you want to use $\alpha$ instead of $a$?
> > - page 4 bottom: "function, suffices" -> "function suffices"
> > - upper half of page 5: "estimate, will" -> "estimate will"
> > - Table 1 (a): Dragonnet (Shi et al., 2019) should fit in one line
> >
> > **Conclusion:**
> > The updated paper adresses many of the raised issues about writing, and in my opinion, the quality is better although some issues still remain. The new version is more clear about novelty, and it appears that I slightly overestimated the theoretical novelty in my initial review. As other reviewers noted, the novelty lies mainly in the design and evaluation of RieszNet and ForestRiesz, which use several techniques from the literature to make them debiased and extend them to general functionals. However, the experiments appear to be well done and with good results for the proposed methods. Since a 7 is not available, I raise my score from 6 to 8.

---

### Decision · Program_Chairs · 2022-01-20

**Decision:**

Reject

**Comment:**

In this paper, the problem of estimating the average of a moment function that depends on an unknown regression function.
It heavily relies on prior papers by e.g. Chernozhukov et al. and the actual novel material consists of making these theoretical
results more practical. Experiments for two practical approaches based on neural networks respectively random forests are also reported.

Initially, the presentation of the paper was heavily criticized by the reviewers, but during the rebuttal phase at least some of the issues were removed. Together with some other improvements this lead to an increased average score. However, it seems fair to say that reading the other papers first, is still kind of necessary.

Despite the still unclear novelty the paper has some merits, which in principle make it acceptable. Compared to the other good papers in my batch, however, it is more incremental and the overall contribution is not as strong. For this reason I vote for rejection, but a comparison to other papers outside my batch is probably a good idea.